# Freeze all-first versus biopsy-first: A retrospective analysis of frozen blastocyst transfer cycles with preimplantation genetic testing for aneuploidy

Eun Jeong Yu[1], Eun-A. Park[2,3], Seung-Ah Choe[1], Kyung-Ah Lee[3], You Shin Kim[1]*

**1** CHA Fertility Center Seoul Station, CHA University, Seoul, Republic of Korea, **2** CHA Fertility Center Seoul Station, Fertility Laboratory, Seoul, Republic of Korea, **3** Department of Biomedical Science, College of Life Science, CHA University, Gyeonggi-do, Republic of Korea

☯ These authors contributed equally to this work.
* medikys@cha.ac.kr

**Data Availability Statement:** All relevant data are within the paper and its Supporting Information files.

## Abstract

Potential use of preimplantation genetic testing for aneuploidy (PGT-A) is increasing. Patients who have excess embryos cryopreserved at the blastocyst stage may desire PGT-A but there is little data available on options for these patients. We compared the efficacy and safety of the timing on the cryopreservation and trophectoderm(TE) biopsy for preimplantation genetic testing for aneuploidy (PGT-A) program associated with the better outcomes after frozen blastocyst transfer. Retrospective analysis of patients who underwent PGT-A cycles from January 2016 to December 2019 was carried out. 2684 blastocysts from cycles were subjected to TE biopsy for performing array comparative genomic hybridization test and Next-generation sequencing. All cycles were divided into two according to the timing of biopsy: biopsy-first (n = 211 cases/ 232 transfers) versus freeze all-first (n = 327 cases/ 415 transfers). In the biopsy-first group, embryos were cultured to expanded blastocyst and proceed to TE biopsy on day 5 or day 6 followed by cryopreservation. In the freeze all-first, blastocysts were vitrified and warmed before biopsy. Rates of clinical pregnancy (52.3% vs. 38.7%, P = 0.09) and ongoing pregnancy (44.3% vs. 34.5%, P = 0.07) in biopsy-first were significantly higher than those in freeze all-first. Biopsy-first showed comparable miscarriage rate with freeze all-first (15.2% (33/217) vs.11.1% (10/90), respectively). Rate ratio (RR) for clinical pregnancy was lower in freeze all-first group (adjusted RR = 0.78, 95% confidence interval: 0.65, 0.93). The RRs for miscarriage and live birth was also lower but it did not reach statistical significance. Our result supported performing TE biopsy of blastocyst for PGT-A before vitrification and warming. This finding would contribute to more evidence-based decision in PGT-A cycles.

## Introduction

Preimplantation genetic testing for aneuploidy (PGT-A) is an evolving technique that improve the effectiveness of assisted reproduction technology treatment in patients at high risk of

**Funding:** The author(s) received no specific funding for this work.

**Competing interests:** The authors have declared that no competing interests exist.

embryonic chromosomal abnormalities, such as advanced maternal age, recurrent miscarriage, and repeated implantation failure [1]. PGT-A, also known before as preimplantation genetic screening (PGS), was first applied more than 20 years ago [2, 3]. Ongoing pregnancy rate per embryo transfer is about 50% in PGT-A cycles [4]. With more evidence supporting the higher success rate of PGT-A, the use of PGT-A in the routine IVF practice is increasing worldwide [5].

Embryo biopsy followed by fresh embryo transfer was traditionally performed in the PGT-A cycle. However, before embryo transfer, the time allowed for genetic analysis of the specimens is limited, particularly after blastocyst biopsy. Cryopreservation of blastocysts after biopsy instead of fresh transfer permits more sufficient time for performance of molecular diagnosis [6]. In addition, cryopreservation of embryos may be beneficial for high responders with risk of ovarian hyperstimulation syndrome or suboptimal endometrium [7].

Although trophectoderm biopsy has become very popular for PGT-A, there is no consensus regarding the timing of trophectoderm biopsy (before versus after cryopreservation) [8]. Some previous studies suggested that the important processes, including embryonic genome activation, genomic imprint maintenance, and methylation reprogramming of non-imprinted genes, occur in the preimplantation stage [9, 10]. It is because the reduction in viable embryonic material and disruption of cell-to-cell communication during biopsy and cryopreservation procedure might play a negative effect on embryo development and clinical outcomes [9]. We hypothesized that biopsy after cryopreservation-thawing may cause the embryo irreparable damage in the freezing process decreasing survival rates and implantation potential. However, previous study is limited by small numbers regarding the success rate of biopsy-first and freeze all-first in frozen ET cycles [11]. This study was to compare the clinical outcomes between biopsy-first and freeze all- first approach in frozen blastocyst transfer cycles combined with PGT-A.

## Material and methods

### Study design

This was a retrospective study using the hospital data of a single fertility center. We included 655 frozen blastocyst ET cycles combined with PGT-A conducted in 538 couples from January 2016 to December 2019. Patients were between the age of 28 and 45 years during IVF and PGT-A cycles. Only the cycles using own oocytes were included in the analysis. Indications for PGT-A were recurrent implantation failure [12, 13], recurrent miscarriage [14, 15], advanced maternal age (≥38 years) [16–18]. The reasons that necessitated vitrification of embryos included development of ovarian hyperstimulation syndrome (OHSS), technical problems encountered during the PGT-A tests, and poor ovarian response that required repeated stimulation and serial vitrification to obtain enough embryos for PGT-A. In addition, surplus embryos from patients with recurrent miscarriages, advanced maternal age, and recurrent implantation failure were vitrified after embryo transfer following IVF without PGT-A. A total of 1,469 blastocysts from 655 cycles were subjected to trophectoderm (TE) biopsy for performing array comparative genomic hybridization (aCGH) test and Next-generation sequencing. We divided final 647 frozen ET cycles into two groups according to the timing of TE biopsy: Freeze all-first (freeze all first and TE biopsy after warming prior to ET) versus Biopsy-first (TE biopsy first and freeze all 'normal' embryos). All patients gave written informed consent for their anonymized medical records to be used for clinical research purpose and the study was approved by the Institutional Review Board of CHA Gangnam Medical Center (Approval No.GCI-18-15).

## Ovarian stimulation protocol

All patients were stimulated with recombinant follicle-stimulating hormone (rFSH) with either an agonist or antagonist protocol. Ovarian response monitoring was performed using serial vaginal ultrasonography. The initial dose of gonadotropin was individualized for each patient according to the woman's age, anti-Mullerian hormone (AMH), basal follicle-stimulating hormone (FSH) levels, antral follicle count (AFC), and previous ovarian response to ovarian stimulation. The daily dose of gonadotropin was adjusted for each individual according to the serum estradiol ($E_2$) concentration, follicular growth and numbers were assessed by ultrasound. When dominant follicles reached 14 mm in mean diameter, 0.25 mg/day of a Gonadotropin-releasing hormone (GnRH) antagonist (Orgalutran®, Organon, Oss, The Netherlands or Cetrotide®; EMD Serono, Rockland, MA, USA) was initiated and was continued until the day of recombinant human chorionic gonadotropin (r-hCG) injection. When at least two follicles with a mean diameter of 18mm were observed, 250 ug or 500 ug of r-hCG (Ovidrel®; Merck, Kenilworth, NJ, USA) or GnRH agonist (Decapeptyl®, Ipsen Pharma, Barcelona, Spain) was injected subcutaneously. Oocyte retrieval was performed 34 to 36 hours after hCG or injection using a 17-gauge needle under transvaginal ultrasonography guidance. All patients had intracytoplasmic sperm injection (ICSI) for insemination because of the possibility of genetic testing of embryos. Fertilization was examined 16–18 h after ICSI and all embryos were cultured to the blastocyst stage. The luteal support was provided for all patients with progesterone vaginal suppositories or progesterone intramuscular injection.

## Trophectoderm biopsy and PGT-A (aCGH) protocol

For TE biopsy, an 18-mm hole was made in the zona pellucida of all embryos on day 4. On day 5 or day 6, trophoblasts that had herniated out of the zona pellucida were chosen for biopsy with a minimum quality >3BC as assessed using Gardner criteria [19]. The blastocysts for TE biopsy were loaded in culture dishes, which contained two to three microdroplets of a blastocyst medium (Sage BioPharma, Inc.) overlaid with paraffin oil (Vitrolife, Kungsbacka, Sweden). The blastocysts were held using a holding pipette (Humagen, Charlottesville, VA, USA), and laser pulses (Zilos TK laser, Hamilton Thorne) were used to punch a small hole in the ZP away from the inner cell mass to accommodate the passage of several TE cells. Approximately 5–10 TE cells detached from the ZP were aspirated into the biopsy pipette pipette (internal diameter, 30um) with smooth suction. The aspirated cells were detached from the blastocysts with several laser pulses combined with smooth suction. The detached cells were aspirated into the TE biopsy pipette and released into the biopsy drop. The TE cells were washed in phosphate-buffered saline solution (PBS-Merk, Germany) and then stored in RNAse—DNAse-free polymerase chain reaction tubes containing 2 μl PBS and were genetically analyzed at the genetic analysis laboratory (Genomecare Inc, Seoul, Korea). On the basis of these results, embryos were classified as either euploid, aneuploid, or no result.

## Embryo vitrification and warming

The biopsied and non-biopsied blastocysts were first equilibrated in a mixture of HEPES medium (SAGE Quinn's-HEPES; CooperSurgical, Trumbull, CT, USA) and 20% HSA (SAGE, CooperSurgical) supplemented with 7.5% ethylene glycol (EG) and 7.5% dimethyl sulfoxide (DMSO; Sigma-Aldrich, St. Louis, MO, USA). For the final equilibration, 15% EG, 15% DMSO and 0.5 M sucrose were used. Each blastocyst was loaded onto a gold electron microscopic (EM) grid (EM Grid; SPI Supplies, West Chester, PA, USA). For the warming process, the EM grid containing the blastocyst was sequentially transferred to culture dishes containing HEPES medium and 0.5 M, 0.25 M, 0.125 M, and 0.0 M sucrose at intervals of 2.5 minutes,

with 20% human serum albumin (SAGE BioPharma). After warming, the blastocyst was washed with blastocyst medium (SAGE In-Vitro Fertilization & Cooper Surgical company, Teumbull, USA) at 37˚C in an atmosphere of 6% $CO_2$, 5% $O_2$ and 89% $N_2$ and then cultured. The vitrification and thawing procedure were performed according to the manufacturer's instructions. During the study period, personnel and protocols related with embryo vitrification and warming in the laboratory were not changed.

## Uterine preparation and embryo transfer (ET)

Once biopsy results confirmed at least one euploid embryo, patients were scheduled for a frozen ET cycle. Most of cycles were performed in hormonal replacement cycles. On the menstrual day 3, we started administering a daily 6 mg of oral estradiol valerate (Progynova®, Schering [Korea] Ltd., Seoul, South Korea). When endometrial thickness reached approximately 8 mm, luteal support was provided in the form of daily vaginal or intramuscular progesterone. One or two euploid embryos were transferred after evaluation of the embryo quality.

In the freeze-all first cycles, thawing blastocysts were prioritized based on the best quality before biopsy. Embryos were warmed in the early morning on the day before transfer, then cultured for 2-3h and assessed for TE biopsy. Biopsy results confirmed on the morning of scheduled transfer and transferred at least one euploid embryo. In biopsy-first cycles, once biopsy results confirmed at least one euploid embryo, patients were scheduled for a frozen ET cycle.

Only those blastocysts that survived the thawing and reexpansion process were considered suitable for transfer. Blastocysts were considered to have survived verification-warming if > 75% of cells were intact after warming. The transfer procedure was performed under transvaginal ultrasound guidance.

## Follow-up and outcomes measured

Embryo survival rate was calculated by dividing the number of embryo that were viable after warming and before embryo transfer by the total number of frozen embryos (from 0% survival for no viable embryo to 100% survival when all embryos were viable). Clinical pregnancy was diagnosed when a gestational sac with fetal heartbeat is present at the 6-week ultrasound. Live birth rate (LBR) was defined as a fetus born alive beyond the 24 weeks of pregnancy.

## Statistical analysis

Student's t-test and Mann-Whitney test were used to compare the difference between the groups. Covariates included women's age at oocyte retrieval. Anti-Mullerian hormone (AMH), follicle-stimulating hormone (FSH), body mass index (BMI), infertility duration, number of previous IVF cycles, number of oocyte retrieval, number of euploid embryo transferred, euploid rate per embryo. We calculated adjusted rate ratio (RR) using log-binomial regression analysis. The analyses were performed using R (ver. 3.6.2; R Development Core Team, Vienna, Austria).

## Results

The number of developed blastocysts is 2684 embryos in 647 cycles. A total of 211 patients were treated with freeze all-first protocol. 327 patients were treated with PGT-A protocols with biopsy-first (Table 1). Women's age was not significantly different between freeze all-first and biopsy-first groups (37.0 ± 3.9 vs 36.7 ± 4.1 years, P = 0.23). There was no significant difference between the two groups in BMI, basal FSH levels, and PGT-A indications. Live birth rate was higher in the biopsy-first group than in the freeze-all first group with no statistically

**Table 1. Clinical characteristics of women with preimplantation genetic testing for aneuploidy in freeze all-first (freeze all first and biopsy) versus biopsy-first (biopsy first and freeze normal).**

|  | Freeze all-first | Biopsy-first | P value |
|---|---|---|---|
| **No. of patients** | 211 | 327 | - |
| **No. of ET cycles** | 232 | 415 | - |
| **Women's age (years)** | 37.0±3.9 | 36.7±4.1 | 0.23 |
| **AMH (ng/ml)** | 4.3±3.3 | 6.0±3.4 | 0.31 |
| **FSH (mIU/ml)** | 7.8±2.6 | 7.3±2.6 | 0.15 |
| **BMI (kg/m$^2$)** | 21.3±2.9 | 21.8±3.1 | 0.27 |
| **Infertility duration (years)** | 4.5±2.9 | 4.0±2.7 | 0.42 |
| **No. of prior IVF cycles** | 2.7±1.7 | 3.1±2.9 | 0.10 |
| **No. of oocyte retrieval** | 18.5±9.0 | 20.8±9.9 | 0.23 |
| **Endometrial thickness on ET (cm)** | 0.96±0.2 | 1.01±0.6 | 0.19 |
| **No. of euploid embryo transferred** | 1.3±0.4 | 1.5±0.7 | 0.34 |
| **Euploidy rate (%)** | 39.9±35.4 | 46.4±21.1 | 0.23 |
| **PGT-A indication (%)** |  |  |  |
| Advanced age (>40 year) | 125(59.2%) | 222(67.8%) | 0.48 |
| ≥ 3 unexplained recurrent pregnancy losses | 51(24.1%) | 95(29.1%) | 0.61 |
| ≥ 3 recurrent implantation failures | 38(18.0%) | 63(19.2%) | 0.98 |
| Abortus chromosome abnormality | 25(11.8%) | 43(13.1%) | 0.72 |
| >1 combined factor | 67(31.7%) | 112(34.3%) | 0.32 |
| *Mean survival rate of embryos (%) | 98.7±7.7 | 98.3±8.2 | 0.10 |
| **Clinical pregnancy rate per ET** | 90/232(38.7%) | 217/415(52.3%) | 0.09 |
| **Live birth rate per ET** | 80/232(34.5%) | 184/415(44.3%) | 0.07 |
| **Miscarriage rate** | 10/90(11.1%) | 33/217(15.2%) | 0.40 |

Values are presented as mean ± standard deviation or n (%). PGT-A, preimplantation genetic testing for aneuploidy; AMH, anti-Müllerian hormone; FSH, follicle stimulating hormone; BMI, body mass index; ET, embryo transfer; IVF in vitro fertilization

Continuous values are presented as mean ± standard deviation. Frequencies are shown as proportions.

*Survival rate = (the number of embryo that were viable after thawing and before embryo transfer/total number of frozen embryos) x100

significance (34.5 vs. 44.3%, *P* = 0.07). The multiple pregnancy rate in the freeze-all first group was 5% (all dizygotic twins), the rate was not significantly different in the biopsy-first group, 3% (all dizygotic twins). No obstetrical complications, such as preeclampsia, placenta abruption, placenta previa and intrauterine growth restriction, were seen between freeze all-first and biopsy-first groups.

The average proportion of euploid ones in biopsied embryos between two groups was similar (39.9 ± 35.4 vs 46.4 ± 21.1, p = 0.23). Rates of clinical pregnancy (38.7% vs 52.3%, P = 0.09) and live birth rate (34.5% vs 44.3%, P = 0.07) in biopsy-first were significantly higher compared to those in freeze all-first (Fig 1). Adjusted RR for clinical pregnancy was 0.78 (95% CI: 0.65–0.93) when conversing freeze all-first versus biopsy-first. The RRs for miscarriage (0.74, 95% CI: 0.38, 1.42) and for live birth (0.89, 95% CI: 0.67, 1.17) were lower in freeze all-first group but not statistically significant (Table 2).

## Discussion

We observed lower clinical pregnancy rate in freeze all-first compared to biopsy-first group in frozen cycles conducted with embryo biopsy for PGT-A. The difference in the risk of

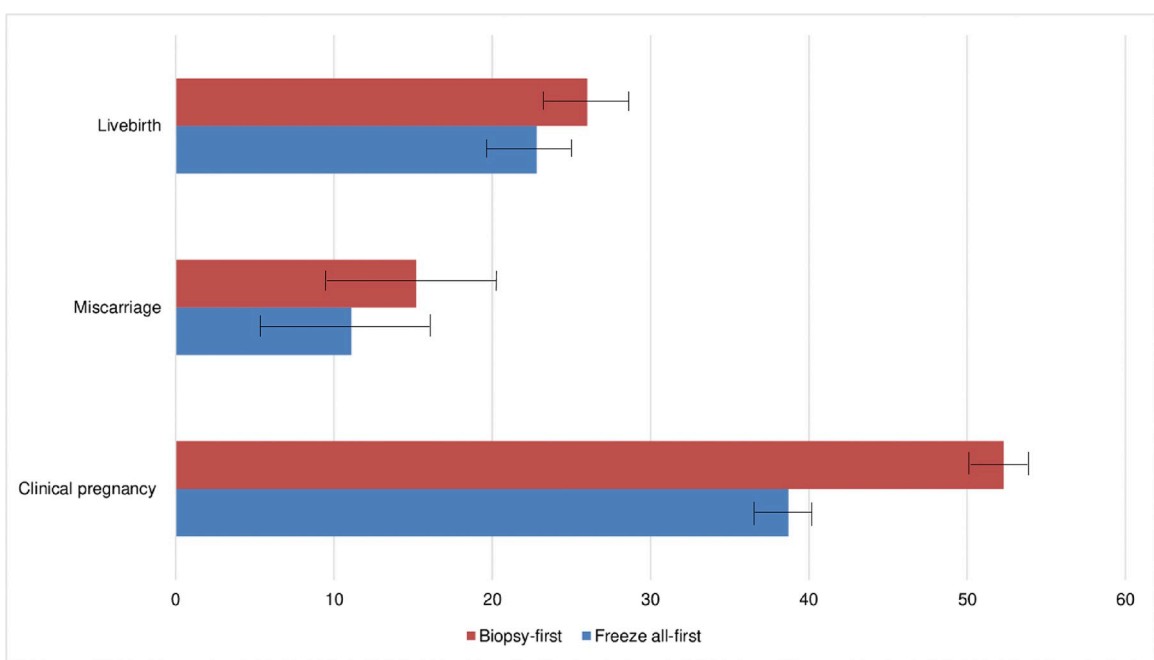

**Fig 1. Crude proportions of clinical pregnancy, miscarriage, and live birth in freeze all-first and biopsy-first cycles.**

miscarriage and live birth were not statistically significant. Our finding supports biopsy-first strategy would be better when frozen blastocyst transfer combined with PGT-A is planned. More recently, Chen et al [20] compared the implantation rates in four groups of blastocysts clustered according to two criteria: vitrified before or later than 3 hours after biopsy, and still collapsed or already re-expanded when the vitrification procedure was initiated. The authors claimed that the blastocysts re-expanding and vitrified later than 3h from biopsy show the highest chance to implant. Lastly, Reed et al [21] in a study designed to compare different cryo-preservation protocols reported no difference in the cryo-survival rate between biopsied and non-biopsied blastocysts. We found that there were no significant differences in embryo survival rates between freeze all-first and biopsy-first groups. It was speculated that cryopreservation, biopsy procedure and their timing in frozen embryo transfer cycle for PGT-A were not associated with increased adverse clinical outcomes.

Defining the optimal time to allow accurate identification of the genetic errors for PGT-A analysis requires careful consideration of several factors [22]. In order not to lose precious euploid blastocysts after warming, also an excellent vitrification program is required. In this regard, several papers in literature reported no blastocyst degeneration after biopsy [23–25] and a survival rate after warming always higher than 95% [23, 26, 27]. In our institution, we also have a vast experience in applying the cryotop method for vitrification [28, 29].

**Table 2. Relative risk (RR) of each pregnancy outcome in freeze all-first (freeze all first and biopsy) versus biopsy-first (biopsy first and freeze normal).**

| | Unadjusted RR (95% CI) | P value | Adjusted RR (95% CI) | P value |
|---|---|---|---|---|
| **Clinical pregnancy** | 0.75 (0.63, 0.91) | 0.00 | 0.78 (0.65, 0.93) | 0.00 |
| **Miscarriage** | 0.76 (0.39, 1.48) | 0.42 | 0.74 (0.38, 1.42) | 0.36 |
| **Live birth** | 0.88 (0.66, 1.17) | 0.37 | 0.89 (0.67, 1.17) | 0.40 |

CI, confidence interval.

Cryopreservation and biopsy procedure may have a negative effect on embryo development potential and decrease the implantation rate [30, 31]. Blastocysts that have been biopsied prior to vitrification already have a hole in the zona pellucida. This hole allows for the direct exposure of cells to cryoprotectant, which may affect survival after warming and implantation [32]. Previous study suggested that injury to the zona pellucida during embryo biopsy makes it more susceptible to damages caused by cryopreservation and thawing [11]. In contrast with these studies, our study showed that biopsied embryos after vitrification and warming has similar survival rate compared to biopsied embryos prior to cryopreservation. Such observations support the finding of NH Zech et al [33]. They reported that the presence of a large opening in the zona pellucida of blastocysts has no negative influence on the survival and further development after vitrification. This evidence, in addition to supporting the reliability and safety of vitrification, further suggests that human blastocysts are resistant to several sources of stress (e.g. manipulations required for IVF) [34, 35].

Interestingly, we observed that biopsied embryo before cryopreservation has more hatched blastocysts (data not shown). Hatching status could make significant different in clinical outcomes between biopsy-first and freeze all first groups. It was also previously reported that embryonic expansion and hatching frequently observed in biopsied embryo, even when they were previously unhatched at time of vitrification [33]. Blastocyst hatching is an important step in the sequence of physiologic events that end up in implantation [36]. Previous studies demonstrated that the small hole that must be created in the zona for embryo biopsy after thawing may also help the embryo to hatch [37]. Although hatching effect on clinical outcomes remains unknown, some studies hypothesized that a fully hatching embryo was more friable and less likely to implant that a non-fully hatching embryo [38]. In a recent study, however, the hatching status was not associated with implantation, clinical pregnancy, and live birth [39]. Also, this change has been related with increased risk of monozygotic twin because of blstocyst herniation and embryo splitting through non-natural gap [40].

This study needs caution in interpretation. As a retrospective study, residual factors might have confounded the association between biopsy timing and pregnancy. For example, although the two groups were comparable for baseline clinical characteristics in general, FSH and BMI of the two groups were different which might have led the difference in the IVF outcomes. To confirm our findings, prospective randomization clinical trial to assess the impact of timing of biopsy would be necessary.

## Conclusion

Our results indicate that PGT-A first in frozen blastocyst transfer cycles is good at clinical outcomes. Based on our finding, we recommend that biopsy-first strategy in frozen ET cycles which is conducted with PGT-A to optimize the IVF outcome.

## Supporting information

**S1 File.**
(XLSX)

## Acknowledgments

The authors would like to thank our laboratory colleagues for providing the vitrification, thawing and PGT-A results. We also thank the clinicians of CHA fertility center Seoul station for their practice.

## Author Contributions

**Conceptualization:** Eun Jeong Yu, Eun-A. Park, Seung-Ah Choe, Kyung-Ah Lee, You Shin Kim.

**Data curation:** Eun Jeong Yu, Eun-A. Park, Seung-Ah Choe, You Shin Kim.

**Formal analysis:** Eun Jeong Yu, Eun-A. Park, Seung-Ah Choe, You Shin Kim.

**Investigation:** Eun Jeong Yu.

**Writing – original draft:** Eun Jeong Yu, Seung-Ah Choe.

**Writing – review & editing:** Eun Jeong Yu, Eun-A. Park, Seung-Ah Choe, Kyung-Ah Lee, You Shin Kim.

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
