## [Decision Letter · Decision Letter 0]

10 Sep 2020

PONE-D-20-23557

Freeze all-first versus biopsy-first: a retrospective analysis of frozen blastocyst transfer cycles with preimplantation genetic testing for aneuploidy

PLOS ONE

Dear Dr. Kim,

Thank you for submitting your manuscript to PLOS ONE. After careful consideration, we feel that it has merit but does not fully meet PLOS ONE’s publication criteria as it currently stands. Therefore, we invite you to submit a revised version of the manuscript that addresses the points raised during the review process.

This paper performs a new analysis and is therefore an interesting contribution. However both reviewers raise serious concerns towards the rationale and the methodology which dampen the enthusiasm for publication. The authors must deal with these reservations and provide valid answers and most likely additional datasets to reach a level of validity acceptable for publication.

We look forward to receiving your revised manuscript.

Kind regards,

Stefan Schlatt

Academic Editor

PLOS ONE

Journal Requirements:

Reviewers' comments:

Reviewer's Responses to Questions

**Comments to the Author**

1. Is the manuscript technically sound, and do the data support the conclusions?

Reviewer #1: Yes

Reviewer #2: No

2. Has the statistical analysis been performed appropriately and rigorously? 

Reviewer #1: Yes

Reviewer #2: I Don't Know

3. Have the authors made all data underlying the findings in their manuscript fully available?

Reviewer #1: Yes

Reviewer #2: Yes

4. Is the manuscript presented in an intelligible fashion and written in standard English?

Reviewer #1: No

Reviewer #2: Yes

5. Review Comments to the Author

Reviewer #1: The authors of this manuscript have undertaken the task of analyzing retrospectively a considerable data set, in which two protocols in a single center dealing with the sequence of events in PGT-A were compared: (1) first freeze all blastocyst embryos, then thaw, then biopsy (n=211 couples, 232 blastocysts), or (2, the more common approach) first biopsy, then freeze all biopsied embryos (n=327 couples, 415 blastocysts). One or two embryos were transferred. The outcome of both protocols was analyzed up to ongoing pregnancy (12th week of gestation). Pregnancy outcome data were not provided. The authors conclude that the sequence of events, in which the embryos are first biopsied and then stored frozen is beneficial to the outcome of the treatment, as given by the ongoing pregnancy rate.

Having searched the literature, this is the first report dealing with this question. Given the clear-cut beneficial outcome of first biopsy the embryo before cryostorage, it is unlikely that a randomized control trial will ever be carried out (although suggested by the authors at the end of the discussion).

Both groups were not completely similar, as the basal FSH levels were higher in the first group, together with a somewhat higher mean BMI in the second group. It would certainly be appropriate to add some details on the decision processes in each of both study groups, perhaps integrated into Table 1 or in a separate Table.

The patients were treated in the time interval from Januar 2016 to Dezember 2019, but the follow-up included only ongoing pregnancies until week 12 of gestation. Most pregnancies must now have gone to delivery. Isn’t it possible to add all information about the outcome of the pregnancies, including the occurrence of complications during pregnancy and the status of the newborns?

The paper would benefit a lot from a thorough and careful edit by a native English speaker.

The authors might consider adding one reference (Cimadomo D, Rienzi L, Romanelli V, et al. Inconclusive chromosomal assessment after blastocyst biopsy: prevalence, causative factors and outcomes after re-biopsy and re-vitrification. A multicenter experience. Hum Reprod 2018;33:1839-1846), in which embryos with inconclusive results of genetic testing were subjected to thawing and re-biopsy. They provide some information about the feasibility of the (re)-biopsy of thawed embryos and their competence to become implanted after embryo transfer, as studied in this retrospective data set.

Figure 1: the authors should design the figure such, that the confidence intervals are visible.

Table 1: Do the authors provide us with standard deviations, standard errors or 95% confidence intervals?

Line 74: replace the word “inevitable” by “irreparable”.

Line 97: all patients

Line 126: zona pellucida

Line 239: instead of (data not seen) please write (data not shown)

Discussion, lines 238 to 248: the authors might add some information about the influence of making a hole into the zona pellucida and the risk of monozygotic twinning.

Reviewer #2: Freeze all-first versus biopsy-first: a retrospective analysis of frozen blastocyst transfer cycles with preimplantation genetic testing for aneuploidy

Manuscript by Jeong Yu et al., retrospectively compared the efficacy and safety of the timing on the cryopreservation and trophectoderm (TE) biopsy for PGT-A program. The main observation is increased clinical and ongoing pregnancy rates in biopsy-first approach than those in freeze all-first. Authors believe that performing TE biopsy of blastocyst for PGT-A before vitrification and warming can give better clinical outcome.

Major comments:

My major concern with this manuscript is lack of methodological details and importantly, rationale in comparing these two groups. There is no clarity in their objective. Even after reading the manuscript several times, I did not understand when embryo transfers were done in freeze- all first group, post warming and TE biopsy. Are the results available fast enough to perform ET in the same cycle, within a few hours? Or biopsied embryos were re-frozen? The methodology is not clear on this point. If re-freezing was done, then the survival rate and embryo quality of re-frozen/thawed embryos needs to be stated, and discussion on the safety of multiple freeze/thaw would be warranted. Also, if the embryos were re-frozen, the survival rate of ~98% in both groups is very interesting.

The author states, "We hypothesized that biopsy before cryopreservation may cause the embryo inevitable damage in the freezing process decreasing survival rates and implantation potential." Is it in comparison to biopsy after cryopreservation? The hypothesis has not been addressed in discussion, especially since the conclusion is contrary to the hypothesis.

Was ICSI done for the freeze-all group too, when PGT was not planned at that point in time? If not, what measures were taken to avoid contamination during IVF to prevent false results in PGT.

Table 1: Mean age of patients is stated as 37.0±3.9 for freeze all first and 36.7±4.1 for biopsy first, but under indications for PGT, advanced age(>40yrs) shows 125(59.2%) and 222(67.8%), respectively. There appears to be some discrepancy in the data which needs to be addressed.

Minor comments:

1. Line 67 - "there is no consensus regarding the timing of TE biopsy (before versus after cryopreservation)". Please cite reference.

2. Line 76 - does the "previous data" refer to the earlier reference? It is not clear.

3. Ln 65-66 - reference given for this sentence has not been cited properly under references.

4. Ln 87 - " old age" can be rephrased as advanced maternal age and cut off can be mentioned. If any criteria were followed for the indications, please cite the references.

5. Line 127 - Blastocyst morphology ">3BC". The criteria used for grading can be stated; was 3BC the minimum grading to be eligible for biopsy? If so, it can be stated accordingly to avoid confusion.

6. Line 131 - Details could be given regarding how the authors proceeded with the biopsy material, with a focus on duration of genetic analysis.

7. Line 140 - Embryo transfer : How long after biopsy were the embryos transferred would require mentioning in the methodology section.

8. Line 149/150 - "in biopsy-first cycles, where the blastocyst had not been subjected to assist hatching in the previous cycle, assisted hatching was performed" - are the authors referring to the freeze-all group? It's not clear.

9. Table 1 : Mean survival rate - It can be stated as mean survival rate of embryos

10. Line 238 - 242 not comprehensible in discussion, could be rephrased for better clarity.

11. A further discussion on pregnancy outcomes & miscarriage rates between the two groups and also in comparing the findings with other studies would be warranted, since the study is looking into safety and efficacy of PGT timing.

12. Spelling mistakes in a few places that need correction.

6. PLOS authors have the option to publish the peer review history of their article (what does this mean?). If published, this will include your full peer review and any attached files.

Reviewer #1: No

Reviewer #2: No

---

## [Author Response · Author response to Decision Letter 0]

2 Nov 2021

Reviewer #1: The authors of this manuscript have undertaken the task of analyzing retrospectively a considerable data set, in which two protocols in a single center dealing with the sequence of events in PGT-A were compared: (1) first freeze all blastocyst embryos, then thaw, then biopsy (n=211 couples, 232 blastocysts), or (2, the more common approach) first biopsy, then freeze all biopsied embryos (n=327 couples, 415 blastocysts). One or two embryos were transferred. The outcome of both protocols was analyzed up to ongoing pregnancy (12th week of gestation). Pregnancy outcome data were not provided. The authors conclude that the sequence of events, in which the embryos are first biopsied and then stored frozen is beneficial to the outcome of the treatment, as given by the ongoing pregnancy rate.

Having searched the literature, this is the first report dealing with this question. Given the clear-cut beneficial outcome of first biopsy the embryo before cryostorage, it is unlikely that a randomized control trial will ever be carried out (although suggested by the authors at the end of the discussion).

Both groups were not completely similar, as the basal FSH levels were higher in the first group, together with a somewhat higher mean BMI in the second group. It would certainly be appropriate to add some details on the decision processes in each of both study groups, perhaps integrated into Table 1 or in a separate Table.

The patients were treated in the time interval from Januar 2016 to Dezember 2019, but the follow-up included only ongoing pregnancies until week 12 of gestation. Most pregnancies must now have gone to delivery. Isn’t it possible to add all information about the outcome of the pregnancies, including the occurrence of complications during pregnancy and the status of the newborns?

-> We added your point to the table 1 and result section 

The paper would benefit a lot from a thorough and careful edit by a native English speaker.

->We received English correction. 

The authors might consider adding one reference (Cimadomo D, Rienzi L, Romanelli V, et al. Inconclusive chromosomal assessment after blastocyst biopsy: prevalence, causative factors and outcomes after re-biopsy and re-vitrification. A multicenter experience. Hum Reprod 2018;33:1839-1846), in which embryos with inconclusive results of genetic testing were subjected to thawing and re-biopsy. They provide some information about the feasibility of the (re)-biopsy of thawed embryos and their competence to become implanted after embryo transfer, as studied in this retrospective data set.

->Thanks for your suggestion. We added this article on result and reference section. 

Figure 1: the authors should design the figure such, that the confidence intervals are visible.

->We added the confidence intervals on figure.

Table 1: Do the authors provide us with standard deviations, standard errors or 95% confidence intervals?

->We have corrected your point.

Line 74: replace the word “inevitable” by “irreparable”.

->We have corrected your point.

Line 97: all patients

->We have corrected your point.

Line 126: zona pellucida

->We have corrected your point.

Line 239: instead of (data not seen) please write (data not shown)

->We have corrected your point.

Discussion, lines 238 to 248: the authors might add some information about the influence of making a hole into the zona pellucida and the risk of monozygotic twinning.

-> we corrected what was pointed out

“Also, this change has been related with increased risk of monozygotic twin because of blstocyst herniation and embryo splitting through non-natural gap”(Line 279-281)

Reviewer #2: Freeze all-first versus biopsy-first: a retrospective analysis of frozen blastocyst transfer cycles with preimplantation genetic testing for aneuploidy

Manuscript by Jeong Yu et al., retrospectively compared the efficacy and safety of the timing on the cryopreservation and trophectoderm (TE) biopsy for PGT-A program. The main observation is increased clinical and ongoing pregnancy rates in biopsy-first approach than those in freeze all-first. Authors believe that performing TE biopsy of blastocyst for PGT-A before vitrification and warming can give better clinical outcome.

Major comments:

My major concern with this manuscript is lack of methodological details and importantly, rationale in comparing these two groups. There is no clarity in their objective. Even after reading the manuscript several times, I did not understand when embryo transfers were done in freeze- all first group, post warming and TE biopsy. Are the results available fast enough to perform ET in the same cycle, within a few hours? Or biopsied embryos were re-frozen? The methodology is not clear on this point. If re-freezing was done, then the survival rate and embryo quality of re-frozen/thawed embryos needs to be stated, and discussion on the safety of multiple freeze/thaw would be warranted. Also, if the embryos were re-frozen, the survival rate of ~98% in both groups is very interesting.

->Thanks for your comment. We tried to explain about this point on material and method part.

In the freeze-all first cycles, thawing blastocysts were prioritized based on the best quality before biopsy. Embryos were warmed in the early morning on the day before transfer, then cultured for 2-3h and assessed for TE biopsy. Biopsy results confirmed on the morning of scheduled transfer and transferred at least one euploid embryo. In biopsy-first cycles, once biopsy results confirmed at least one euploid embryo, patients were scheduled for a frozen ET cycle.(Line 169-174)

The author states, "We hypothesized that biopsy before cryopreservation may cause the embryo inevitable damage in the freezing process decreasing survival rates and implantation potential." Is it in comparison to biopsy after cryopreservation? The hypothesis has not been addressed in discussion, especially since the conclusion is contrary to the hypothesis.

->Thanks for your comment. We are sorry for confusion. We changed the paragraph from “biopsy before cryopreservation” to “biopsy after cryopreservation-thawing”(Line 75)

Was ICSI done for the freeze-all group too, when PGT was not planned at that point in time? If not, what measures were taken to avoid contamination during IVF to prevent false results in PGT.

->We used ICSI in all cases. ICSI is commonly used in pre-implantation genetic testing (PGT) cases to eliminate any risk of sperm DNA contamination.

Table 1: Mean age of patients is stated as 37.0±3.9 for freeze all first and 36.7±4.1 for biopsy first, but under indications for PGT, advanced age(>40yrs) shows 125(59.2%) and 222(67.8%), respectively. There appears to be some discrepancy in the data which needs to be addressed.

->Freeze all first group included more young patient with recurrent spontaneous abortion than biopsy first group.

Minor comments:

1. Line 67 - "there is no consensus regarding the timing of TE biopsy (before versus after cryopreservation)". Please cite reference.

-> We have corrected your point.

Reference 8 : Aoyama N, Kato K. Trophectoderm biopsy for preimplantation genetic test and technical tips: A review. Reprod Med Biol. 2020;19(3):222-31.

2. Line 76 - does the "previous data" refer to the earlier reference? It is not clear.

->We changed the word from “data” to “study”. This refers to an article published prior to our study.

3. Ln 65-66 - reference given for this sentence has not been cited properly under references.

->We found new reference and change the reference

Greco E, Litwicka K, Minasi MG, Cursio E, Greco PF, Barillari P. Preimplantation Genetic Testing: Where We Are Today. Int J Mol Sci. 2020;21(12).

3. Ln 87 - " old age" can be rephrased as advanced maternal age and cut off can be mentioned. If any criteria were followed for the indications, please cite the references.

-> We have corrected your point and added the references. (line 89)

4. Line 127 - Blastocyst morphology ">3BC". The criteria used for grading can be stated; was 3BC the minimum grading to be eligible for biopsy? If so, it can be stated accordingly to avoid confusion.

-> We newly described about this point at the material and method part (line 128)

6. Line 131 - Details could be given regarding how the authors proceeded with the biopsy material, with a focus on duration of genetic analysis.

-> We newly described about this point at the material and method part (line 128-140)

7. Line 140 - Embryo transfer : How long after biopsy were the embryos transferred would require mentioning in the methodology section.

-> We newly described about this point at the material and method part (line 169-174)

8. Line 149/150 - "in biopsy-first cycles, where the blastocyst had not been subjected to assist hatching in the previous cycle, assisted hatching was performed" - are the authors referring to the freeze-all group? It's not clear.

-> This sentence corresponds to biopsy-first group. 

9. Table 1 : Mean survival rate - It can be stated as mean survival rate of embryos

-> Thanks for your comment. We have corrected your point.

10. Line 238 - 242 not comprehensible in discussion, could be rephrased for better clarity.

-> Thanks for your comment. We have corrected your point.

“Hatching status could make significant different in clinical outcomes between biopsy-first and freeze all frist groups.”

11. A further discussion on pregnancy outcomes & miscarriage rates between the two groups and also in comparing the findings with other studies would be warranted, since the study is looking into safety and efficacy of PGT timing.

->Thanks for your comment. We have corrected your point.

12. Spelling mistakes in a few places that need correction.

-> Thanks for your comment. We have corrected your point.

---

## [Decision Letter · Decision Letter 1]

26 Nov 2021

PONE-D-20-23557R1Freeze all-first versus biopsy-first: a retrospective analysis of frozen blastocyst transfer cycles with preimplantation genetic testing for aneuploidyPLOS ONE

Dear Dr. Kim,

Thank you for submitting your manuscript to PLOS ONE. After careful consideration, we feel that it has merit but does not fully meet PLOS ONE’s publication criteria as it currently stands. Therefore, we invite you to submit a revised version of the manuscript that addresses the points raised during the review process. The authors have addressed the concerns of the reviewers in a not fully adequate way. There are still doubts on the validity of the conclusions. The major concerns is that patient selection is responsible for some of the observed effects. Before a final decision can be made the authors have to deal once more with the critical concerns and add additional evidence (and supportive data) that their conclusions are truly valid.

We look forward to receiving your revised manuscript.

Kind regards,

Stefan Schlatt

Academic Editor

PLOS ONE

Reviewers' comments:

Reviewer's Responses to Questions

**Comments to the Author**

1. If the authors have adequately addressed your comments raised in a previous round of review and you feel that this manuscript is now acceptable for publication, you may indicate that here to bypass the “Comments to the Author” section, enter your conflict of interest statement in the “Confidential to Editor” section, and submit your "Accept" recommendation.

Reviewer #1: (No Response)

Reviewer #2: All comments have been addressed

2. Is the manuscript technically sound, and do the data support the conclusions?

Reviewer #1: No

Reviewer #2: Yes

3. Has the statistical analysis been performed appropriately and rigorously? 

Reviewer #1: N/A

Reviewer #2: Yes

4. Have the authors made all data underlying the findings in their manuscript fully available?

Reviewer #1: No

Reviewer #2: Yes

5. Is the manuscript presented in an intelligible fashion and written in standard English?

Reviewer #1: No

Reviewer #2: Yes

6. Review Comments to the Author

Reviewer #1: The authors of this manuscript evaluate the effect of a freeze-all first strategy versus a biopsy-first strategy on the outcome of PGT-A in a cohort of more than 500 treatment cycles. The analysis was retrospective, but the strength of this study is that no other study has dealt with this topic so far. The paper has already undergone a major revision based up two extensive and high quality reviews. The quality of the manuscript has much improved, but a number of issues still remain.

As this is a retrospective analysis, it is important to explain why both strategies were used (freeze-all first versus biopsy-first). Between lines 87 and 93 some information is given, but not quantified. Especially the statement that poor ovarian reserve requiring repeated stimulations as a reason for freeze-all first is worrying, as this subpopulation is characterized by lower likelihood of pregnancy, thereby skewing the outcome of the study. This would also explain the slightly higher (but statistically significant) basal FSH-serum levels in the one study group. The authors should clearly state and quantify the reasons for selection into the one or into the other study group, either in the Material and Method section and/or in the Results section of their manuscript, as already requested by Reviewer 2.

One reviewer requested the authors to add live birth numbers to their results section. This was indeed partially done, but the authors should also implement Table 1 with these additional data. They should add the live birth numbers to the clinical pregnancy rate per ongoing ET, to ongoing pregnancy rate per ET and to miscarriage. The first reviewer also asked for data on incident complications during pregnancy and data on the newborn babies, but these data were not given (although stated to be given). Please, amend this valuable information, that will greatly benefit the quality of this study.

The authors should also clarify in Table 1, whether the mean values are accompanied by standard error of the mean or by standard deviations.

The authors have transferred either one or two euploid embryos. Twin pregnancies may have resulted. In addition, the hole inserted into the zona pellucida of the blastocyst embryos has had an effect on the timing of the hatching, as stated in the discussion. For that reason, the authors should also include information about the multiple pregnancies (including whether these were monozygotic or dizygotic twin pregnancies).

Table 2 should also be implemented with an estimate of the miscarriage rate, as PGT-A may lower the incidence of miscarriages. It is of interest, whether the miscarriage rate is indeed different in one of both strategies.

Figure 1 should also have some indication of the statistical analysis (such as p-values). As the reviewers requested: confidence intervals were given.

One statement in the Introduction is misleading to me: why should the timing of cryopreservation and biopsy in the blastocyst stage interfere with genome activation and epigenetic imprinting? I would transfer lines 64 to 67 to line 57, after the first sentence. This sequence of statements is more logical.

Although languagewise the manuscript has been improved, the first few sentences of the discussion must be improved. For example:

Line 29: Abstract: the sentence should be: “Potential use of preimplantation genetic testing for aneuploidy (PGT-A) is increasing.”

Line 126: zona pellucida instead of zona pellucid.

Line 207: lower pregancy rates

Line 239: data not shown instead of data not seen

Line 241: frequently often: remove one of both words

Reviewer #2: Authors have addressed my comments in the revised version and I do not have any other issues with the manuscript.

7. PLOS authors have the option to publish the peer review history of their article (what does this mean?). If published, this will include your full peer review and any attached files.

Reviewer #1: No

Reviewer #2: No

---

## [Author Response · Author response to Decision Letter 1]

31 Mar 2022

Reviewer #1: The authors of this manuscript evaluate the effect of a freeze-all first strategy versus a biopsy-first strategy on the outcome of PGT-A in a cohort of more than 500 treatment cycles. The analysis was retrospective, but the strength of this study is that no other study has dealt with this topic so far. The paper has already undergone a major revision based up two extensive and high quality reviews. The quality of the manuscript has much improved, but a number of issues still remain.

As this is a retrospective analysis, it is important to explain why both strategies were used (freeze-all first versus biopsy-first). Between lines 87 and 93 some information is given, but not quantified. Especially the statement that poor ovarian reserve requiring repeated stimulations as a reason for freeze-all first is worrying, as this subpopulation is characterized by lower likelihood of pregnancy, thereby skewing the outcome of the study. This would also explain the slightly higher (but statistically significant) basal FSH-serum levels in the one study group. The authors should clearly state and quantify the reasons for selection into the one or into the other study group, either in the Material and Method section and/or in the Results section of their manuscript, as already requested by Reviewer 2.

We added your point to the material and methods (line 89-94). As a result of reviewing the raw data again, there was no statistically significant difference between FSH and BMI. 

One reviewer requested the authors to add live birth numbers to their results section. This was indeed partially done, but the authors should also implement Table 1 with these additional data. They should add the live birth numbers to the clinical pregnancy rate per ongoing ET, to ongoing pregnancy rate per ET and to miscarriage. The first reviewer also asked for data on incident complications during pregnancy and data on the newborn babies, but these data were not given (although stated to be given). Please, amend this valuable information, that will greatly benefit the quality of this study.

We added your point to the table 1 and result section

The authors should also clarify in Table 1, whether the mean values are accompanied by standard error of the mean or by standard deviations.

We added your point to the table 1 and result section

The authors have transferred either one or two euploid embryos. Twin pregnancies may have resulted. In addition, the hole inserted into the zona pellucida of the blastocyst embryos has had an effect on the timing of the hatching, as stated in the discussion. For that reason, the authors should also include information about the multiple pregnancies (including whether these were monozygotic or dizygotic twin pregnancies).

We added your point to the result section

Table 2 should also be implemented with an estimate of the miscarriage rate, as PGT-A may lower the incidence of miscarriages. It is of interest, whether the miscarriage rate is indeed different in one of both strategies.

We changed the term from ‘spontaneous abortion’ to ‘miscarriage’ in table 2

Figure 1 should also have some indication of the statistical analysis (such as p-values). As the reviewers requested: confidence intervals were given.

We have corrected your point

One statement in the Introduction is misleading to me: why should the timing of cryopreservation and biopsy in the blastocyst stage interfere with genome activation and epigenetic imprinting? I would transfer lines 64 to 67 to line 57, after the first sentence. This sequence of statements is more logical.

We have corrected your point

Although languagewise the manuscript has been improved, the first few sentences of the discussion must be improved. For example:

Line 29: Abstract: the sentence should be: “Potential use of preimplantation genetic testing for aneuploidy (PGT-A) is increasing.”

Line 126: zona pellucida instead of zona pellucid.

Line 207: lower pregancy rates

Line 239: data not shown instead of data not seen

Line 241: frequently often: remove one of both words

We have corrected your point.

Reviewer #2: Authors have addressed my comments in the revised version and I do not have any other issues with the manuscript.

---

## [Editor Report · Decision Letter 2]

13 Apr 2022

Freeze all-first versus biopsy-first: a retrospective analysis of frozen blastocyst transfer cycles with preimplantation genetic testing for aneuploidy

PONE-D-20-23557R2

Dear Dr. Kim,

We’re pleased to inform you that your manuscript has been judged scientifically suitable for publication and will be formally accepted for publication once it meets all outstanding technical requirements.

Kind regards,

Stefan Schlatt

Academic Editor

PLOS ONE

Additional Editor Comments (optional):

The authors responded appropriately to the reviewers concerns and suggestions. The paper has much improved, contains valid and original datasets which will be important for the field.
---

## [Editor Report · Acceptance letter]

18 Sep 2022

PONE-D-20-23557R2 

Freeze all-first versus biopsy-first: a retrospective analysis of frozen blastocyst transfer cycles with preimplantation genetic testing for aneuploidy  

Dear Dr. Kim:

I'm pleased to inform you that your manuscript has been deemed suitable for publication in PLOS ONE. Congratulations! Your manuscript is now with our production department. 

Kind regards, 

on behalf of

Dr. Stefan Schlatt 

Academic Editor

PLOS ONE